# A Framework for Toxic PFAS Replacement based on GFlowNet and Chemical Foundation Model

**Eduardo Soares**
IBM Research Brazil
Rio de Janeiro, RJ, Brazil
`eduardo.soares@ibm.com`

**Flaviu Cipcigan**
IBM Research Europe
Warrington, United Kingdom
`flaviu.cipcigan@ibm.com`

**Dmitry Zubarev**
IBM Research Almaden
San Jose, CA, USA
`dmitry.zubarev@ibm.com`

**Emilio Vital Brazil**
IBM Research Brazil
Rio de Janeiro, RJ, Brazil
`evital@br.ibm.com`

## Abstract

Per- and polyfluoroalkyl substances (PFAS) are a broad class of molecules used in almost every sector of industry and consumer goods. PFAS exhibit highly desirable properties such as high durability, water repellance or high acidity, that are difficult to match. As a side effect, PFAS persist in the environment and have detrimental effect on human health. Epidemiological research has linked PFAS exposure to chronic health conditions, including dyslipidemia, cardiometabolic disorders, liver damage, and hypercholesterolemia. Recently, public health agencies significantly strengthed regulations on the use of PFAS. Therefore, alternatives are needed to maintain the pace of technological developments in multiple areas that traditionally relied on PFAS. To support the discovery of alternatives, we introduce MatGFN-PFAS, an AI system that generates PFAS replacements. We build MatGFN-PFAS using Generative Flow Networks (GFlowNets) for generation and a Chemical Language Model (MolFormer) for property prediction. We evaluate MatGFN-PFAS by exploring potential replacements of PFAS superacids, defined as molecules with negative pKa, that are critical for the semiconductor industry. It might be challenging to eliminate PFAS superacids entirely as a class due to the strong constraints on their functional performance. The proposed approach aims to account for this possibility and enables the generation of safer PFAS superacids as well. We evaluate two design strategies: 1) Using Tversky similarity to design molecules similar to a target PFAS and 2) Directly generating molecules with negative pKa and low toxicity. In this paper, we studied 6 PFAS molecules that have the structure defined as $R - CF_2OCF_2 - R'$. For the given query PFAS SMILE **CC1CC(CC(F)(F)C(F)(F)OC(F)(F)C(F)(F)S(=O)(=O)O)OC1=O**, MatGFN-PFAS system was able to generate a candidate with very low toxicity, $LD50 = 7304.23$, strong acidity, $pKa = -1.92$, and high similarity score, $89.32\%$, to the studied PFAS molecule. Results demonstrated that the proposed MatGFN-PFAS was able to consistently generate replacement molecules following all the constraints forehead mentioned. The resulting datasets for this ongoing study are available at `https://ibm.box.com/v/MatGFN-PFAS-generated-datasets`.

## 1 Introduction

Per- and polyfluoroalkyl substances (PFAS) belong to a category of compounds within the broader domain of organofluorine substances [1]. In 2021, the Organisation for Economic Cooperation

37th Conference on Neural Information Processing Systems (NeurIPS 2023).

and Development (OECD) introduced an updated definition for PFAS [2], characterizing them as fluorinated substances that incorporate at least one fully fluorinated methyl or methylene carbon atom, devoid of any hydrogen ($H$), chlorine ($Cl$), bromine ($Br$), or iodine ($I$) atoms bound to it. PFAS have demonstrated their utility across a diverse spectrum of consumer goods and industrial applications [3].

While PFAS offer exceptional advantages in the production of both consumer and industrial items, their robustness and durability also contribute to their remarkable resistance to degradation [4]. Consequently, numerous older-generation PFAS have accumulated in the environment, wildlife, and human beings over time [5, 6, 7]. As a result, a blend of legacy PFAS, as well as newer alternatives and emerging variants, persistently pervades our surroundings, notably including global food and water supplies [8, 9]. Furthermore, the bioaccumulation of PFAS in human blood has been widely documented, with detection in approximately 98% of adult Americans [10].

Given the extensive bioaccumulation and persistence characteristics of PFAS, the field of epidemiological research focusing on the adverse health impacts of PFAS exposure in humans is rapidly expanding [11]. A multitude of epidemiological investigations has identified associations between PFAS exposure and chronic ailments such as dyslipidemia and cardiometabolic disorders [12, 13, 14, 15]. Human studies have also established connections between elevated serum PFAS levels and liver damage [16, 17], including increased alanine aminotransferase (ALT) levels [18], steatosis [19], and non-alcoholic fatty liver disease (NAFLD) severity [14]. Furthermore, epidemiological revealed links between PFAS exposure and elevated cholesterol and triglyceride levels [20, 21].

Stated the significant adverse impacts on both ecological and human health associated with PFAS substances, there is an urgent need for research aimed at identifying potential alternatives to mitigate these harmful effects [22]. This paper introduces MatGFN-PFAS, an AI system that harnesses the power of the recently developed Generative Flow Network (GFlowNet) [23]. In this particular study, we employ MoLFormer [24], a large chemical language model, to predict the toxicity (LD50) [25] and pKa of the generated molecules, which serves as the basis for the GFlowNet's reward function. Furthermore, we incorporate Tversky similarity within the reward function to generate molecules with a structural resemblance to the PFAS molecules currently under investigation [26].

## 2  Methodology

In this section, we explain the methodological framework of MatGFN-PFAS delineated within this study. As depicted in Figure 1, we present an intricately devised schema for the generation of candidates to replace PFAS substances leveraging GFlowNet. This approach takes into account three key components: the MoLFormer-derived toxicity and pKA prediction for the synthesized molecule and the Tversky similarity metric quantifying the resemblance between the target molecule (the entity to be replaced) and the synthesized molecule, weighing shared structures. These constituents are systematically employed as components of a reward function, aiming the generation of optimal candidate molecules for the PFAS replacement endeavor.

### 2.1  GFlowNet

The GFlowNet algorithm is designed to learn a generative policy that constructs objects by following a sequence of actions, as detailed in the comprehensive description provided by [23]. This policy operates within a user-defined deterministic Markov decision process (MDP) [27]. The MDP encompasses a state space denoted as $S$, a set of permissible actions $A_s$ associated with each state $s$, a deterministic transition function $S \times A_s \rightarrow S$, and a reward function $R$. GFlowNets conceptualize this MDP as a directed graph referred to as a flow network. In this representation, states correspond to nodes, and the MDP's transition function defines directed edges between these nodes. A state's children are states reachable through outgoing edges, while its parents are the sources of incoming edges. States lacking outgoing edges are termed terminal states or sinks and are denoted as $x \in X$. It is essential for GFlowNets that users define the MDP in a way that ensures this graph remains acyclic, contains precisely one state without incoming edges, known as the initial state (source), and adheres to $R : X \rightarrow \mathbb{R} \geq 0$.

A complete trajectory is a sequence of states $\tau(s_0 \rightarrow s_1 \rightarrow \cdots \rightarrow s_n)$ starting from the source $s_0$ and leading to a sink $s_n$, with $(s_t \rightarrow s_{t+1}) \in A_{s_t}$ for all $t$. We use $T$ to denote the set of all

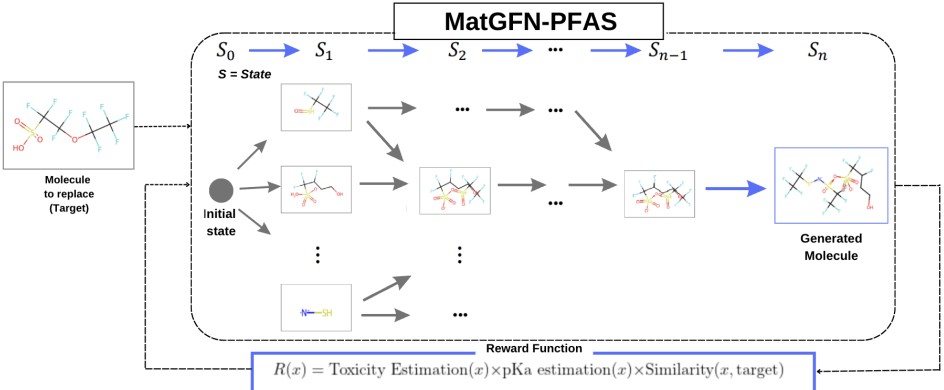

Figure 1: General architecture of the proposed MatGFN-PFAS approach.

complete trajectories. A trajectory flow is a non-negative function $F : T \rightarrow R \geq 0$, describing the unnormalized probability flowing along each complete trajectory $\tau$ from the source to a sink. For any state $s$, the state flow $F(s) = \sum_{\{\tau \in T : s \in T\}} F(\tau)$ quantifies the total unnormalized probability passing through state $s$. Additionally, for any edge $s \rightarrow s'$, a trajectory flow $F(\tau)$ is considered Markovian if there exist distributions $P_F(\cdot|s)$ over the children of every non-terminal state $s$, along with a constant $Z$, such that for any complete trajectory $\tau$, we have $P_F(\tau) = F(\tau)/Z$, and $P_F(\tau = (s_0 \rightarrow s_1 \rightarrow \cdots \rightarrow s_n)) = \prod_{t=1}^{n} = P_F(s_t|s_{t-1})$. This $P_F(s_t|s_{t-1})$ is termed a forward policy, which can be used to sample complete trajectories from $F$.

GFlowNets are trained using stochastic gradient descent to optimize the learning objective on states or trajectories sampled from a training policy [27]. This training policy is typically a combination of $P_F^{\theta}$ and a uniform action policy, serving to encourage exploration during training. In the realm of Reinforcement Learning, GFlowNet training is considered off-policy. Importantly, GFlowNet training is a bootstrapping process where the current policy is utilized to sample new $x$ at each training round. Since $R(x)$ is defined by the user, it is computed for each new $x$, and this set $\{x, R(x)\}$ is employed to update the GFlowNet policy.

In our specific case, $R(x)$ is defined as $R(x) = \text{Toxicity prediction}(x) \times \text{pKa prediction}(x) \times \text{Tversky Similarity}(x, Target)$, with Target representing the PFAS molecule for which we seek a substitute. We use absolute values of pKa, and generated positive pKas are penalized, receiving the value 0.00001. The significance of a negative pKa value lies in ensuring the production of superacids, which find essential applications across various industries such as chip manufacturing [28]. While it may be challenging to eliminate their use entirely [28], the objective of the proposed approach is to generate safer alternatives.

## 2.2   MoLFormer

MoLFormer [24], is a large-scale masked chemical language model that processes inputs through a series of blocks that alternate between self-attention and feed-forward connections. MoLFormer was trained in a self-supervision manner with 1.1 billion molecules from PubChem and ZINC datasets and uses tokenization process, as detailed in [29]. The MoLFormer vocabulary includes 2362 unique chemical tokens. These tokens are used to fine-tune or retrain the MolFormer model. To reduce computation time, the sequence length has been limited to a range of 202 tokens as 99.4% percent of all 1.1 billion molecules contain less than 202 tokens.

MOLFORMER is equipped with a self-attention mechanism that allows the network to construct complex representations that incorporate context from across the sequence of SMILES. By transforming the sequence features into queries ($q$), keys ($k$), and value ($v$) representations, attention mechanisms can weigh the importance of different elements within the sequence. This enables the model to learn highly informative representations of the input data, making it a powerful tool for predicting

molecular properties. In this work, we used a version of the MoLFormer that was trained to predict toxicity, LD50 and pKa of PFAS elements [25]. The model reported 75% of accuracy on the toxicity task, and it is state-of-the-art in this domain.

## 2.3 MatGFN-PFAS training details

The following parameters have been used to train the MatGFN-PFAS for each of the 6 PFAS molecules studied in this experiment:

Table 1: MatGFN-PFAS agent parameters

| Parameter | Value |
|---|---|
| Learning rate | 5e-3 |
| Epochs | 35000 |
| Mini batch size | 5 |

Fig. 2 shows the loss trajectory while training the MatGFN-PFAS to generate molecules based on the studied SMILES **CC1CC(CC(F)(F)C(F)(F)OC(F)(F)C(F)(F)S(=O)(=O)O)OC1=O**, Fig. 2 also shows the logZ for the MatGFN-PFAS training. The other studied SMILES demonstrated similar behavior.

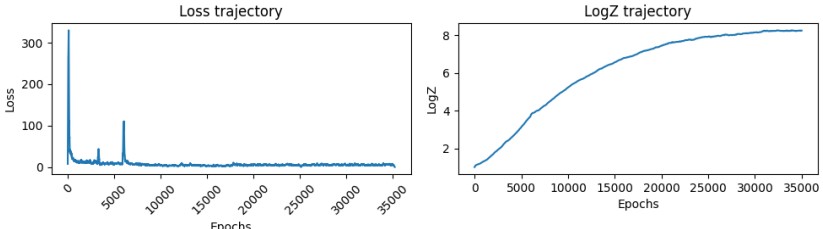

Figure 2: The figure illustrates the loss trajectory for the MatGFN-PFAS training. Losses are stabilized after epoch 6,000. LogZ for the MatGFN-PFAS training is also illustrated.

# 3 Results

To evaluate our proposed approach we selected a set of 6 PFAS SMILES which the LD50 measurement between 50-500 mg/kg, moderate toxicity according to the U.S. Environmental Protection Agency (EPA) [30]. The studied PFAS molecules have the structure defined as $R - CF_2OCF_2 - R'$, where $R$ and $R'$ can either be $F$, $O$, or saturated carbons. This is a PFAS definition stated by the U.S EPA agency [31]. Table 2 provides details about the studied PFAS molecules.

Table 2: Studied PFAS molecules which have the structure as $R - CF_2OCF_2 - R'$

| SMILES | LD50 (mg/kg body weight) | pKA |
|---|---|---|
| CC1CC(CC(F)(F)C(F)(F)OC(F)(F)C(F)(F)S(=O)(=O)O)OC1=O | 416.0 | -3.12 |
| O=C1CCC(CC(F)(F)C(F)(F)OC(F)(F)C(F)(F)S(=O)(=O)O)O1 | 406.0 | -3.12 |
| O=S(=O)(O)C(F)(F)C(F)(F)OC(F)(F)C(F)(F)C(F)(F)C(F)(F)OC(F)(F)C(F)(F)S(=O)(=O)O | 272.0 | -2.87 |
| O=S(=O)(O)C(F)(F)C(F)(F)OC(F)(F)C(F)(F)C(F)(F)F | 236.0 | -3.22 |
| O=S(=O)(O)C(F)(F)C(F)(F)OC(F)(F)C(F)(F)C1CC2CCC1C2 | 423.0 | -3.12 |
| O=S(=O)(O)C(F)(F)C(F)(F)OC(F)(F)C(F)(F)F | 416.0 | -3.25 |

Figure 3, illustrates the top 5 results in terms of Tversky similarity to the query molecule: **CC1CC(CC(F)(F)C(F)(F)OC(F)(F)C(F)(F)S(=O)(=O)O)OC1=O**. It is possible to note through the preliminary results that the proposed algorithm tries to generate molecules with low toxicity and pKa at the same time that it tries to maximize the similarity between the generated molecule and the target molecule.

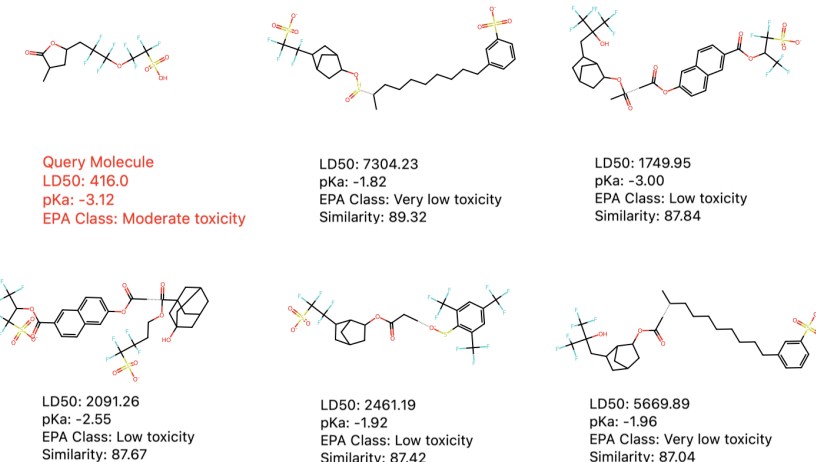

Figure 3: Top 5 generated molecules in terms of similarity score to the query SMILES: **CC1CC(CC(F)(F)C(F)(F)OC(F)(F)C(F)(F)S(=O)(=O)O)OC1=O**.

Initial results demonstrate that our algorithm exhibits a dual focus. Firstly, it strives to produce molecules with reduced toxicity and pKa levels, aligning with the goal of enhancing drug safety and minimizing potential side effects. Secondly, it places a strong emphasis on maximizing the similarity between the generated molecule and the target molecule. This dual objective highlights the algorithm's versatility and its potential to contribute significantly to drug discovery and design processes. By achieving a delicate balance between toxicity reduction and structural similarity. For the other studied molecules MatGFN-PFAS also generated toxic PFAS replacement candidates with low toxicity, low pKa, and similarity scores ranging from $85\%$ to $95\%$. Detailed results for the other studied PFAS SMILES are included in the Supplementary Materials. The resulting datasets of this ongoing work are available at `https://ibm.box.com/v/MatGFN-PFAS-generated-datasets`.

The reported findings constitute an ongoing research effort, necessitating future investigations to validate the proposed framework's robustness. Emphasis is placed on testing the framework across an expanded and more complex PFAS dataset, encompassing diverse compounds to assess adaptability. Additionally, there is a call for the development of rigorous mechanisms to evaluate the quality of generated molecules, incorporating advanced criteria aligned with PFAS toxicity, structural fidelity, and environmental impact. These proposed avenues aim to refine the framework, positioning it as an effective tool for the generation of environmentally sustainable alternatives to toxic PFAS compounds.

## 4   Conclusion

In this paper, we introduce MatGFN-PFAS, an AI system leveraging GFlowNet and chemical foundation model, MoLFormer, for the purposeful generation of a diverse set of molecular candidates earmarked for the substitution of toxic per- and polyfluoroalkyl substances (PFAS). The architectural foundation of MatGFN-PFAS integrates a nuanced reward function, incorporating predictive indices such as LD50 and pKa, alongside the Tversky similarity metric. Our ongoing investigation substantiates the efficacy of this method, elucidating its proficiency in yielding molecules characterized by minimized toxicity and superacidic pKa values. Notably, MatGFN-PFAS is designed to concurrently optimize both structural fidelity and chemical congruence to the specified PFAS target.

For future research, our paramount objective resides in test the proposed framework on a expanded and more complex PFAS dataset, encompassing diverse compounds to assess adaptability. Furthermore, our commitment extends towards the refinement of the quality metrics pertaining to the generated molecular candidates. Subsequent endeavors are oriented towards the calibration of MatGFN-PFAS, ensuring its capacity not only to yield molecular candidates with commendable toxicity profiles and structural fidelity but also to align with the exacting standards requisite for robust PFAS replacement strategies. The resulting datasets generated by this ongoing investigation can be found at `https://ibm.box.com/v/MatGFN-PFAS-generated-datasets`.

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

# Supplementary Materials

## Additional results

For each PFAS molecule studied in this work, we generated 1000 new replacement candidates. Below we present the top 5 generated molecules, ranked by their similarity scores to the PFAS molecules utilized as queries for MatGFN-PFAS:

1. Fig.4 illustrates the top generated molecules in terms of similarity to the query: **O=C1CCC(CC(F)(F)C(F)(F)OC(F)(F)C(F)(F)S(=O)(=O)O)O1**, $LD50 = 406.0, pKa = -3.12$

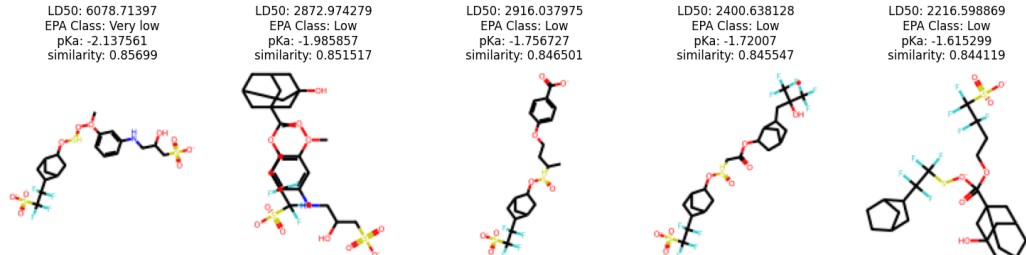

Figure 4: Top 5 results in terms of similarity to the query SMILE: **O=C1CCC(CC(F)(F)C(F)(F)OC(F)(F)C(F)(F)S(=O)(=O)O)O1**.

2. Results for the query: **O=S(=O)(O)C(F)(F)C(F)(F)OC(F)(F)C(F)(F)C(F)(F)C(F)(F)OC(F)(F)C(F)(F)S(=O)(=O)O**, $LD50 = 272.0, pKa = -2.87$, are illustrated by Fig.5.

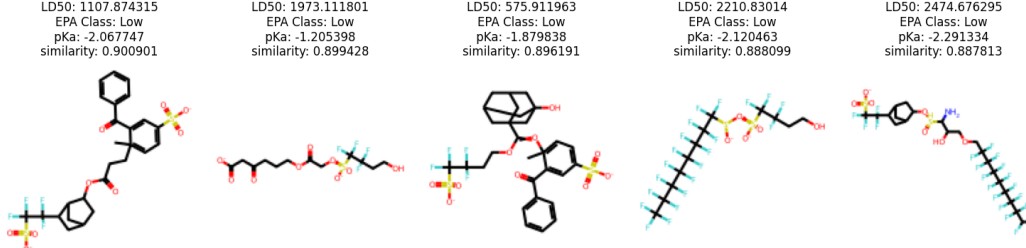

Figure 5: Top 5 results in terms of similarity to the query SMILE:
**O=S(=O)(O)C(F)(F)C(F)(F)OC(F)(F)C(F)(F)C(F)(F)C(F)(F)OC(F)(F)C(F)(F)S(=O)(=O)O**.

3. The top results in terms of similarity to the query: **O=S(=O)(O)C(F)(F)C(F)(F)OC(F)(F)C(F)(F)C(F)(F)F**, $LD50 = 236.0, pKa = -3.22$ are illustrated in Fig.6.

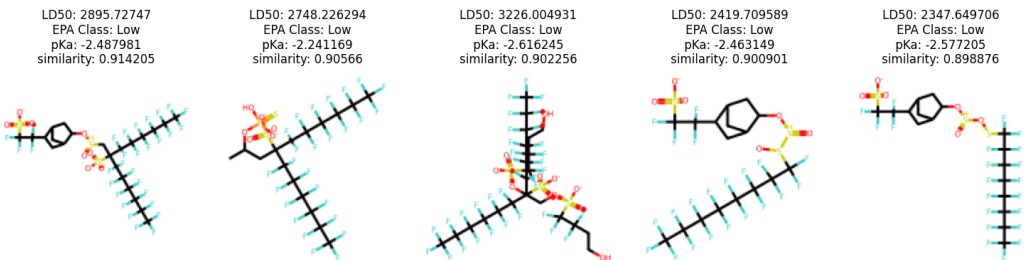

Figure 6: Top 5 results in terms of similarity to the query SMILE:
**O=S(=O)(O)C(F)(F)C(F)(F)OC(F)(F)C(F)(F)C(F)(F)F**.

4. For the query **O=S(=O)(O)C(F)(F)C(F)(F)OC(F)(F)C(F)(F)C1CC2CCC1C2**, $LD50 = 423.0$, $pKa = -3.12$, the top 5 generated results in terms of similarity score are demonstrated by Fig.7.

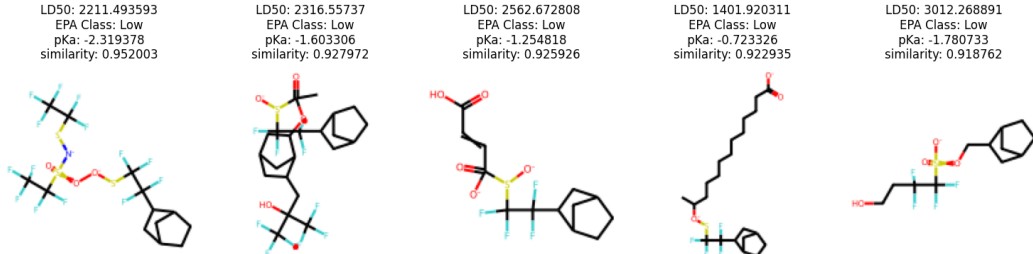

Figure 7: Top 5 results in terms of similarity to the query SMILE:
**O=S(=O)(O)C(F)(F)C(F)(F)OC(F)(F)C(F)(F)C1CC2CCC1C2**.

5. Finally, for the query **O=S(=O)(O)C(F)(F)C(F)(F)OC(F)(F)C(F)(F)F**, $LD50 = 416.0$, $pKa = -3.25$, the results are illustrated by Fig.8.

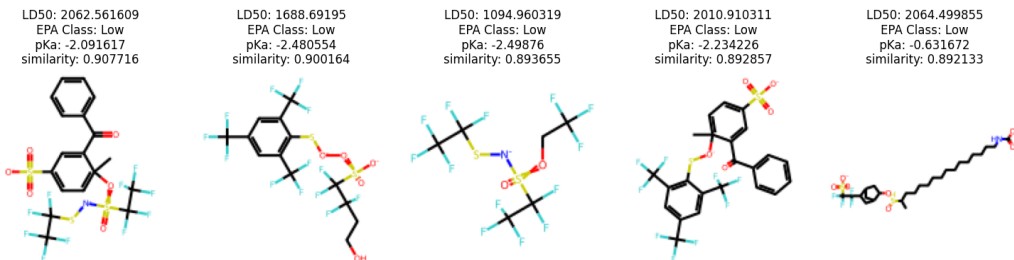

Figure 8: Top 5 results in terms of similarity to the query SMILE:
**O=S(=O)(O)C(F)(F)C(F)(F)OC(F)(F)C(F)(F)F**.

Through these results we can observe that the proposed MatGFN-PFAS framework consistently tries to generate replacement molecules that follows the constraints of having low toxicity in terms of LD50 measurements, negative pKa values (superacids), and high structural similarities to the query molecules, keeping the main the properties of the studied molecules.

These initial findings of this ongoing study demonstrate the effectiveness of the proposed MatGFN-PFAS approach in generating possible toxic PFAS replacements. Future work should concentrate on improving the results as well as evaluating the quality of the generated molecules. The resulting datasets are available at `https://ibm.box.com/v/MatGFN-PFAS-generated-datasets`.

