# OpenReview forum: "A Framework for Toxic PFAS Replacement based on GFlowNet and Chemical Foundation Model"
_NeurIPS.cc/2023/Workshop/AI4Science — NeurIPS2023-AI4Science Poster_

### Official Review · Reviewer_Wcvo · 2023-10-17
**The proposed framework is practical, but the experiments need to be more convincing.**

**Rating:** 5
**Confidence:** 3

**Review:**

Authors discuss about PFAS altenative generation seeking to mitigate its toxic nature by exploring potential alternatives in this insightful study. To this end, the pivotal contribution of the paper lies in the presentation of a comprehensive framework known as MatGFN.  This framework leverages the powerful combination of GflowNet and MoLFormer to generate PFAS molecules and accurately predict their toxicity.

However, it must be noted that the experimental analysis presented in the manuscript falls short of being truly rigorous in showcasing the feasibility of this framework, authors only demonstrate one FPAF alternative generation. Additionally, it is regrettable that the manuscript lacks refinement in certain sections, especially in the results segment. Notably, Table 1 lacks any contextual description, making it challenging for readers to fully comprehend the presented data. Furthermore, Figure 2 appears to have duplicate titles, which may cause confusion among readers seeking a clear understanding of the findings.

An additional observation worth mentioning pertains to the LD50 and pKA values of the first two query SIMILES in Table 1, which are identical. Such sudden repetition raises questions about the accuracy and reliability of the reported data. Moreover, it is puzzling to observe a discrepancy in the LD50 value of CC1CC(CC(F)(F)C(F)(F)OC(F)(F)C(F)(F)S(=O)(=O)O)OC1=O, with Table 1 indicating a value of 406.0, while Figure 2 presents it as 416.0. Such inconsistencies, if left unaddressed, cast a shadow of doubt on the robustness of the reported results.

Finally, one recommendation for the authors is to provide a concise summary of the performance of all query SMILES in the main body of the text. Such a summary would enable readers to grasp the key takeaways of the study more efficiently, allowing for a more comprehensive evaluation of the proposed framework's effectiveness.

In conclusion, despite the notable contributions and potential of the MatGFN framework, the manuscript would greatly benefit from a more rigorous experimental setup, improved clarity in presentation, and enhanced consistency in the reported results.